# Reducing Chemical Fertilizer Application in Greenhouse Vegetable Cultivation under Different Residual Levels of Nutrient

Nannan Zhou [1], Yujiao Chen [1], Jiajia Wang [2], Wenbin Yang [1] and Ying Wang [1,2,3,*]

[1] Collaborative Innovation Center of Recovery and Reconstruction of Degraded Ecosystem in Wanjiang Basin Co-Founded by Anhui Province and Ministry of Education, School of Ecology and Environment, Anhui Normal University, Wuhu 241002, China
[2] Anhui Provincial Key Laboratory of Nutrient Recycling, Resources and Environment, Soil and Fertilizer Research Institute, Anhui Academy of Agricultural Sciences, Hefei 230031, China
[3] Anhui Laboratory of Molecule-Based Materials, Anhui Normal University, Wuhu 241002, China
*   Correspondence: wangying11@ahnu.edu.cn

**Abstract:** Excessive chemical fertilizer application in greenhouse vegetable cultivation results in environmental risks and residual nutrients in the soil. Conventional plot experiments conducted in one field cannot recommend fertilizer reduction to farmers when the residual nutrient levels were various among different fields. In this study, nine plot experiments were simultaneously conducted in nine greenhouse fields where the soils could reflect different residual levels of nutrient, with two application rates (100 and 0%) for each nitrogen, phosphorus, and potassium fertilizer. The results showed that fertilizer reduction did not decrease vegetable yield when soil nitrate, Olsen–phosphorus, and exchangeable potassium were $\geq$173.3, 45.8, and 93.1 mg kg$^{-1}$, respectively. However, no N treatment decreased vegetable yields in fields 1–3 because the inadequately residual nitrate ($\leq$103.9 mg kg$^{-1}$) in these fields led to low nitrate absorbed from sources other than chemical fertilizer, high recovery efficiencies of N, and high productivity of N absorbed from chemical fertilizer. Residual nitrate that soil EC could reflect was the limiting factor of yield under fertilizer reduction. This study indicated that reducing fertilizer in greenhouse vegetable cultivation should be based on the residual level of nutrients in the soil, which is meaningful in agricultural sustainability and environmental safety.

**Keywords:** residual nutrient; fertilizer reduction; stem lettuce; nutrient accumulation; nutrient recovery efficiency





## 1. Introduction

Overuse of chemical fertilizer in vegetable production results in soil deterioration [1] and environmental risks, such as water eutrophication [2] and air pollution [3]. Compared with open-air fields, the suitable days for vegetable production in plastic greenhouse fields are more extended, which generally means more chemical fertilizer application [4]. Therefore, reducing the application of chemical fertilizer in greenhouse vegetable cultivation is essential for the sustainable development of agriculture and the safety of the ecological environment.

A history of excessive application of chemical fertilizers would store nutrients in the soil as mineral nutrients, in roots, immobilized into microorganisms or the other soil organic matter pools; then, they become available for the crops in subsequent cultivating seasons, which is called the residual nutrient effect [5]. Previous literature has confirmed the effect of residual nutrients. For example, a 3–year field experiment found that 14.6–18.7% and 5.4–5.8% of $^{15}$N–labeled fertilizer were recovered by maize in the second and third seasons, respectively [6]. A pot experiment was performed to reveal the residual effect of P

fertilization. The result showed no significant difference in the wheat yield and absorbed P between P and no P treatments [7]. Previous studies that revealed the residual nutrient effect mainly focused on cereal crops. The residual nutrient effect in plastic greenhouse vegetable fields should not be ignored because of the continuously high fertilizer input and largely surplus nutrient.

Application of fertilizer based on farmers' individual experience would result in different residual levels of nutrient in different fields [8,9]. Consequently, the affecting intensities of residual nutrient must be various among different fields. For instance, adequate residual nutrient in greenhouse soils may meet the demands of vegetable production, thus reducing fertilizer application will not affect vegetable yield. Otherwise, it will decrease the yield. Many studies have investigated the responses of vegetable yields to fertilizer reduction in one field divided into several plots [10–13]. Due to the different residual nutrient levels, the recommendation of fertilizer reduction from these studies might only be suitable in the studied field. A fields-scale (on-farm) study should be helpful to clarify the effect of fertilizer reduction on vegetable yield under different residual levels of nutrients, which can give more precision guidelines to local farmers, but related research was limited.

Based on the previous pot experiment [14], we performed nine plot experiments simultaneously in nine fields covered with plastic greenhouses where the soils could reflect different residual nutrient levels in the study area. The present study aims to (i) clarify the responses of vegetable yields to fertilizer reduction under different residual nutrient levels, (ii) determine the critical level of residual nutrients that reducing chemical fertilizer does not decrease vegetable yield, (iii) reveal the affecting mechanism of fertilizer reduction on vegetable yield under the effect of residual nutrient, then find out the limiting factor.

## 2. Materials and Methods

### 2.1. Study Site

The research area was located on the southeastern coast of Dianchi Lake (24°42′ N, 102°42′ E) (Figure 1), which is a typically eutrophic shallow lake in China [15]. The area belongs to a semi-humid region within the subtropical climatic zone. The average annual air temperature is 14.7 °C, ranging from a minimum of −8.1 °C to a maximum of 31 °C. The average annual precipitation averages 782.5 mm, of which more than 90% falls between May and October. The average annual evaporation is 2472.3 mm [16]. A paddy rice-broad bean cropping system with a low nutrient input (210–433 kg N, 0–71 kg P, and 0–123 kg K ha$^{-1}$ y$^{-1}$) was dominant in the last century. After that, paddy rice cultivation has been gradually changed to vegetable cultivation in plastic greenhouses. Greenhouse soils in the area show similar physical properties and dull reddish color. The soil particle size distribution of sand, silt, and clay was 24.0%, 31.6%, and 44.4%, respectively. The cation exchange capacity (CEC) and base saturation percentages were 18.7 cmol$_c$ kg$^{-1}$ and 161.7%, respectively. The contents of exchangeable Ca, Mg, K, and Na were 22.2, 5.52, 1.32, and 0.82 cmol$_c$ kg$^{-1}$, respectively [16]. The contents of iron and aluminum oxides were 38.65–57.09 and 37.47–47.68 g kg$^{-1}$, respectively [17]. The soils can be categorized as Eutric Cambisol (WRB soil classification) [16].

The main crops in the vegetable fields are leafy vegetables, such as stem lettuce (*Lactuca sativa* L.). The growth duration is 30–90 days. The annual cropping intensity ranges from one to seven. Nutrient input from chemical fertilizer varies greatly in different fields (170–2970 kg N, 70–970 kg P, and 100–1870 kg K ha$^{-1}$ year$^{-1}$) [18].

### 2.2. Experimental Design

A large-scale investigation (94 vegetable fields covered with plastic greenhouses) was conducted before the present study. After testing soil nutrients in the 94 vegetable fields, nine fields were carefully selected to conduct the experiment simultaneously from July 2018 to June 2019. The soil nutrient contents in the nine fields could reflect different nutrient levels in the study area (details are shown in Section 3.1) (Figure 1). Considering the workload, each field was divided into 12 plots with 4 treatments and 3 replications for

each treatment. The treatments with different applications of chemical fertilizer are shown in Table 1. Control treatment (N100–P100–K100) reflected the average application level in the research area based on a large–scale investigation. N, P, and K inputs under other treatments were zeroed separately. For example, N0–P100–K100 indicated that not applied. The N, P, and K fertilizers were urea (N ≥ 46.4%), calcium superphosphate ($P_2O_5$ ≥ 16.0%), and potassium chloride (KCl ≥ 95.0%), respectively. Each plot (long × wide: 10.6 m × 5.0 m) was distributed based on a randomized block design.

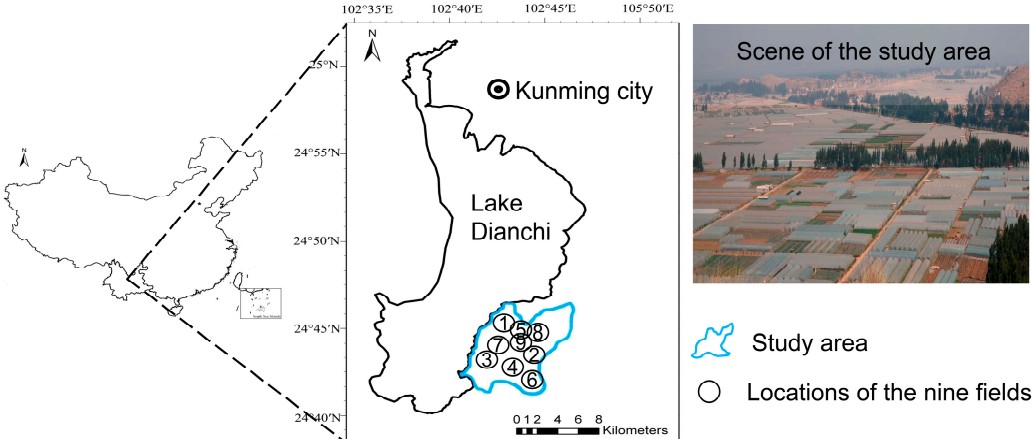

**Figure 1.** Study area and locations of the nine fields. Numbers in the circles represent the fields for the plot experiments.

**Table 1.** Input amounts of nutrients under different treatments.

| Treatment | Nutrient Input (N–P–K kg ha$^{-1}$) |
| --- | --- |
| N100-P100-K100 | 645.3–208–572 |
| N0-P100-K100 | 0–208–572 |
| N100-P0-K100 | 645.3–0–572 |
| N100-P100-K0 | 645.3–208–0 |

Four crops of stem lettuce, a locally common vegetable, were cultivated during the experimental period. Seedlings with 5–6 true leaves showing normal and equal growth status were transplanted in each plot with about 18 plant m$^{-2}$, the average cropping density in the study area. Sprinklers combined with artificial irrigation were utilized to ensure that the soil humidity in each plot was about pF 2.0 (HT-TSW, Huatai, Beijing, China) [19].

*2.3. Soil and Vegetable Sampling and Analysis*

Before the experiment, the surface soils (0–20 cm) were sampled from each field by the "5–point sampling method" after vegetable harvesting to obtain five soil subsamples. Visible crop residues were taken away. The five soil subsamples were thoroughly mixed with a soil drill and were separated into two parts after sieving (<5 mm), one of which was utilized to analyze nitrate content (the main form of N absorbed by vegetables), and the other was air–dried to determine soil total N (TN), total carbon (TC), available P and K, pH, and electric conductivity (EC). Soil nitrate was extracted by 1 mol L$^{-1}$ KCl, then determined by a continuous flow analyzer (San++ System, Skalar Corp., Delft, The Netherlands). Soil TN and TC were analyzed by an elemental analyzer (vario MACRO cube; Elementar Corp., Hanau, Germany). Available soil P (Olsen–P) was extracted by 0.5 mol L$^{-1}$ NaHCO$_3$ and analyzed using the Olsen method [20]. Available soil K was extracted by NH$_4$OAC (1 mol L$^{-1}$) and analyzed using flame photometry (FP6450, Shanghai Yidian Corp., Shanghai, China). Soil EC and pH were analyzed using glass electrodes

(SX751, Sanxin Corp., Shanghai, China) after mixing the soil with distilled water under 1:5 and 1:2.5 soil–to–water ratios, respectively.

The aboveground mature stem lettuce was sampled along the diagonal in each plot. The subsample point area was 1 m$^2$. Each subsample point contained about 18 lettuce plants m$^{-2}$, and the three subsample points were distributed evenly in the area with an average growth condition in each plot. Stem lettuce was dried in an oven (70 °C) to a constant weight, and the total N, P, and K contents were measured according to the methods of Bao [21].

### 2.4. Calculations and Statistical Analysis

N, P, and K in stem lettuce under the treatments N0–P100–K100, N100–P0–K100, and N100–P100–K0 could be considered as N, P, and K uptake from sources other than chemical fertilizer, respectively. The recovery efficiency of applied N was calculated as:

$$\text{REN} = \frac{\text{abN}_{\text{N100–P100–K100}} - \text{abN}_{\text{N0–P100–K100}}}{\text{N}_{100}} \tag{1}$$

N100 was the applied amounts of N from chemical fertilizer under treatments N100–P100–K100. abN$_{\text{N100–P100–K100}}$ and abN$_{\text{N0–P100–K100}}$ were N absorbed by stem lettuce under N100–P100–K100 and N0–P100–K100 treatments, respectively [22]. The recovery efficiencies of applied P (REP) and K (REK) were calculated using a similar method.

The productivity of N absorbed from chemical fertilizer indicates the variation of vegetable weight per unit N uptake from chemical fertilizer ($\Delta Y/\Delta \text{abN}$), which can reflect the contribution of N absorbed from chemical fertilizer to the vegetable yield. In this study, the productivity of N absorbed from chemical fertilizer was calculated as:

$$\Delta Y/\Delta \text{abN} = \frac{Y_{\text{N100–P100–K100}} - Y_{\text{N0–P100–K100}}}{\text{abN}_{\text{N100–P100–K100}} - \text{abN}_{\text{N0–P100–K100}}} \tag{2}$$

Y$_{\text{N100–P100–K100}}$ and Y$_{\text{N0–P100–K100}}$ are the stem lettuce yields under the N100–P100–K100 and N0–P100–K100 treatments, respectively. The productivities of P($\Delta Y/\Delta \text{abP}$) and K($\Delta Y/\Delta \text{abK}$) absorbed from chemical fertilizer were calculated using a similar method [14].

The significant differences between control and no fertilizer treatments were analyzed by independent samples *t*-test. The Pearson correlation was calculated to explore the relationships between different variables. All of the analyses were performed using SPSS 16.0.

### 3. Results

#### 3.1. Soils Properties in the Nine Fields

Soil-available nutrients (nitrate, Olsen–P, and NH$_4$OAC–K) in fields 1–3 showed relatively low contents compared with those in fields 4–9 (Table 2). Soil nitrate and NH$_4$OAC–K in field 9 were the highest among these fields. The relatively high CV of nitrate, Olsen–P, and NH$_4$OAC–K suggested that the selected nine fields could reflect different residual nutrient levels in the study area. Nitrate was positively correlated with EC (Figure 2a) and was negatively correlated with pH (Figure 2b).

**Table 2.** Soil properties in the nine fields.

| Field | TN (g kg$^{-1}$) | TC (g kg$^{-1}$) | Nitrate (mg kg$^{-1}$) | Olsen–P (mg kg$^{-1}$) | NH$_4$OAC–K (mg kg$^{-1}$) | EC (mS m$^{-1}$) | pH |
|---|---|---|---|---|---|---|---|
| 1 | 2.5 | 23.6 | 50.1 | 45.8 | 110.2 | 35.3 | 6.8 |
| 2 | 2.1 | 27.8 | 70.4 | 40.3 | 155.6 | 58.2 | 6.9 |
| 3 | 2.2 | 18.2 | 103.9 | 30.1 | 93.1 | 53.4 | 6.9 |
| 4 | 2.6 | 16.2 | 173.3 | 62.8 | 160.5 | 57.4 | 6.5 |
| 5 | 1.9 | 15.4 | 193.4 | 89.6 | 178.9 | 88.5 | 6.8 |

**Table 2.** *Cont.*

| Field | TN (g kg$^{-1}$) | TC (g kg$^{-1}$) | Nitrate (mg kg$^{-1}$) | Olsen–P (mg kg$^{-1}$) | NH$_4$OAC–K (mg kg$^{-1}$) | EC (mS m$^{-1}$) | pH |
|---|---|---|---|---|---|---|---|
| 6 | 3.0 | 24.4 | 224.5 | 221.2 | 186.2 | 97.6 | 6.7 |
| 7 | 2.7 | 19.2 | 270.9 | 110.4 | 190.5 | 112.7 | 6.5 |
| 8 | 2.6 | 23.0 | 290.3 | 190.5 | 210.4 | 130.9 | 6.5 |
| 9 | 2.8 | 19.1 | 323.2 | 97.4 | 249.1 | 154.2 | 6.3 |
| CV (%) | 14.3 | 19.9 | 52.0 | 67.8 | 28.2 | 45.5 | 3.2 |

CV: Coefficient of variation.

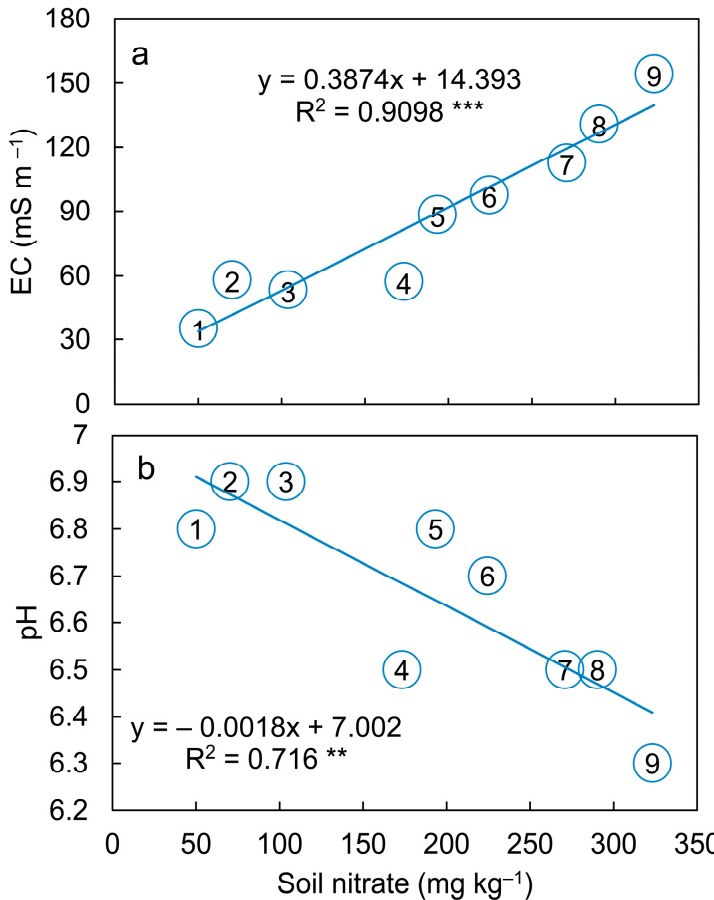

**Figure 2.** Correlations of soil nitrate with soil EC (**a**) or soil pH (**b**). Numbers in the circles represent the fields for the plot experiments. *** significant at $p < 0.001$, ** significant at $p < 0.01$.

### 3.2. Nutrient Absorption and Vegetable Yield under Fertilizer Reduction

Compared with 100% N application, N absorption by stem lettuce under 0% N application significantly decreased by 66.2% in field 1, 79.6% in field 2, 73.7% in field 3, and 17.5% in field 4, respectively, resulting in relatively low N absorption in fields 1–4 (Table 3). For fields 5–9, non–application of N fertilizer did not significantly affect N absorption. P absorption in fields 2 and 3 under 0% P treatment significantly decreased by 44.9% and 36.5% compared with those under 100% P treatment, resulting in relatively low P absorption in the two fields. For other fields, P absorption did not significantly decrease under non-application of P fertilizer. No significant difference was observed between 100% and 0% K treatments for K absorption in all fields. The results indicated that fertilizer reduction resulted in different responses of nutrient (N and P) absorption in different fields.

**Table 3.** Nitrogen, phosphorus, and potassium are absorbed by stem lettuce under different chemical fertilizer applications.

| Treatment | Nutrient Absorbed by Vegetable (kg ha$^{-1}$ y$^{-1}$) | | | | | | | | |
|---|---|---|---|---|---|---|---|---|---|
| | 1 | 2 | 3 | 4 | 5 | 6 | 7 | 8 | 9 |
| | N | | | | | | | | |
| N100-P100-K100 | 428.2 [a] | 402.9 [a] | 504.2 [a] | 525.5 [a] | 535.3 [a] | 465.8 [a] | 595.7 [a] | 534.5 [a] | 513.3 [a] |
| N0-P100-K100 | 144.7 [b] | 82.1 [b] | 132.6 [b] | 433.5 [b] | 506.9 [a] | 439.2 [a] | 583.7 [a] | 491.7 [a] | 479.4 [a] |
| | P | | | | | | | | |
| N100-P100-K100 | 108.1 [a] | 115.7 [a] | 104.2 [a] | 149.6 [a] | 170.1 [a] | 217.1 [a] | 184.7 [a] | 199.6 [a] | 205.9 [a] |
| N100-P0-K100 | 102.1 [a] | 63.7 [b] | 66.2 [b] | 139.7 [a] | 165.8 [a] | 205.6 [a] | 175.1 [a] | 178.9 [a] | 185.9 [a] |
| | K | | | | | | | | |
| N100-P100-K100 | 273.2 [a] | 263.5 [a] | 302.8 [a] | 297.2 [a] | 274.5 [a] | 382.5 [a] | 431.1 [a] | 456.0 [a] | 437.2 [a] |
| N100-P100-K0 | 257.1 [a] | 230.7 [a] | 281.3 [a] | 262.3 [a] | 260.9 [a] | 355.9 [a] | 418.8 [a] | 424.7 [a] | 425.5 [a] |

Different letters within a column indicate significant differences ($p < 0.05$).

Compared with 100% N application, the stem lettuce yield under 0% N application significantly decreased by 73.6% in field 1, 63.7% in field 2, and 49.9% in field 3, respectively (Table 4). For other fields, non-application of N fertilizer did not significantly affect the yield. Reducing P application to zero significantly decreased the yield by 20.8% in field 2 and 31.8% in field 3, respectively, while it did not affect the yield in other fields. Compared with 100% K application, the yield in all fields did not significantly decrease without K fertilizer input. The results indicated that reducing fertilizer decreased vegetable productivity in several fields.

**Table 4.** Stem lettuce yields under different chemical fertilizer applications.

| Treatment | Vegetable Yield (t ha$^{-1}$ y$^{-1}$) | | | | | | | | |
|---|---|---|---|---|---|---|---|---|---|
| | 1 | 2 | 3 | 4 | 5 | 6 | 7 | 8 | 9 |
| N100-P100-K100 | 395.1 [a] | 355.0 [a] | 430.6 [a] | 460.9 [a] | 427.1 [a] | 494.3 [a] | 509.9 [a] | 518.1 [a] | 511.7 [a] |
| N0-P100-K100 | 104.2 [b] | 128.7 [b] | 215.8 [b] | 432.3 [a] | 415.3 [a] | 482.5 [a] | 505.2 [a] | 516.9 [a] | 510.0 [a] |
| N100-P100-K100 | 395.1 [a] | 355.0 [a] | 430.6 [a] | 460.9 [a] | 427.1 [a] | 494.3 [a] | 509.9 [a] | 518.1 [a] | 511.7 [a] |
| N100-P0-K100 | 384.3 [a] | 281.3 [b] | 293.5 [b] | 458.1 [a] | 412.5 [a] | 481.1 [a] | 484.9 [a] | 506.3 [a] | 508.9 [a] |
| N100-P100-K100 | 395.1 [a] | 355.0 [a] | 430.6 [a] | 460.9 [a] | 427.1 [a] | 494.3 [a] | 509.9 [a] | 518.1 [a] | 511.7 [a] |
| N100-P100-K0 | 366.5 [a] | 337.7 [a] | 406.3 [a] | 455.5 [a] | 402.0 [a] | 477.1 [a] | 497.2 [a] | 492.4 [a] | 505.3 [a] |

Different letters within a column indicate significant differences ($p < 0.05$).

*3.3. Correlation of Nutrient Absorbed from Sources Other Than Chemical Fertilizer with Other Variables*

In this study, N, P, and K absorbed from sources other than chemical fertilizer were positively correlated with the soil nitrate, Olsen–P, and NH$_4$OAC–K, respectively (Figure 3a–c). The result suggested that nutrients absorbed from sources other than chemical fertilizer by stem lettuce were mainly from available soil nutrients, namely, residual nutrients of the previous chemical fertilizer.

Significant correlations were observed between N absorbed from sources other than chemical fertilizer and REN (Figure 4a). The REN values in fields 4–9 (ranging from 0.02 to 0.14) were obviously lower than those in fields 1–3 (ranging from 0.44 to 0.58). The results suggested that N absorbed from sources other than chemical fertilizer negatively regulated REN, and a high amount of N absorbed from sources other than chemical fertilizer resulted in a low REN. The correlations were insignificant for P and K (Figure 4b,c). All REP and REK values were less than 0.30 and 0.07, respectively.

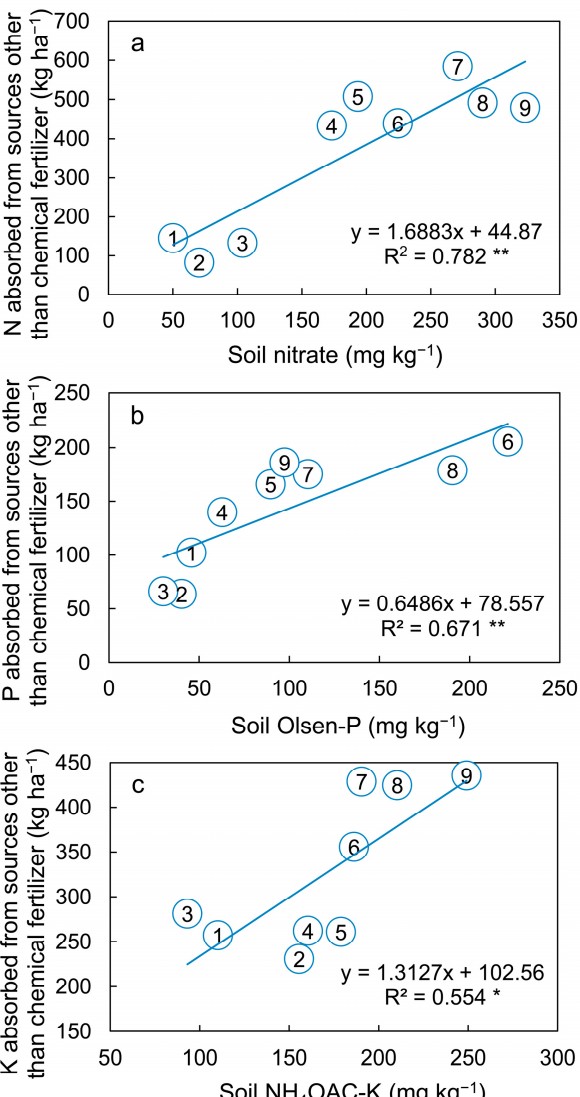

**Figure 3.** Correlation of nutrient absorbed from sources other than chemical fertilizer by stem lettuce with soil nutrients. (**a–c**) Numbers in the circles represent the fields for the plot experiments. ** significant at $p < 0.01$, * significant at $p < 0.05$.

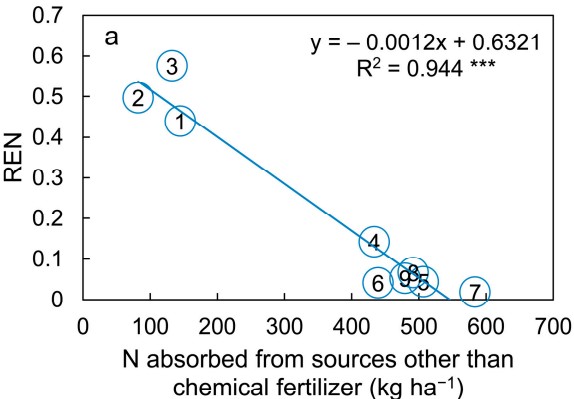

**Figure 4.** *Cont.*

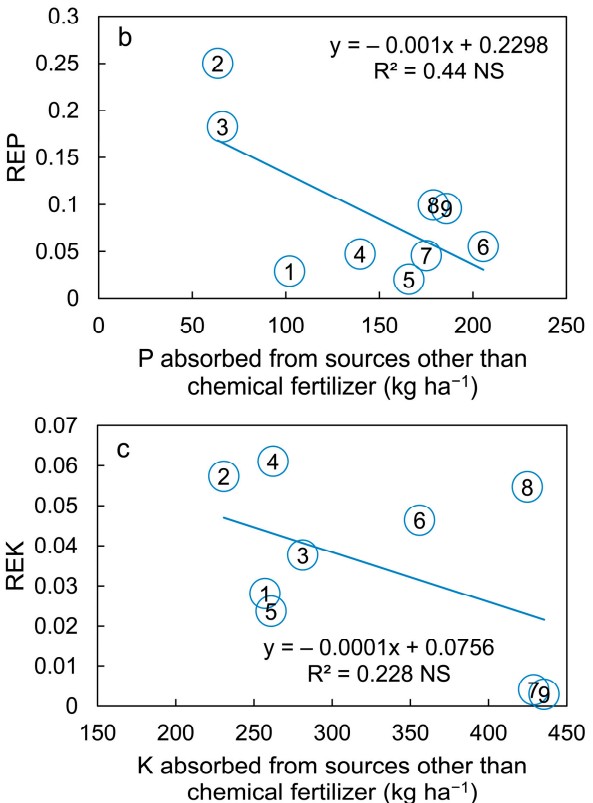

**Figure 4.** Correlation of the recovery efficiency of the applied nutrient using chemical fertilizer with nutrients absorbed from sources other than chemical fertilizer by stem lettuce. REN, REP, and REK represent the recovery efficiencies of applied N, P, and K using chemical fertilizer, respectively. (**a–c**) Numbers in the circles represent the fields for the plot experiments. NS not significant, *** significant at *p* < 0.001.

The productivity of N absorbed from chemical fertilizer was negatively correlated with the N absorbed from sources other than chemical fertilizer; that is, a high amount of N absorbed from sources other than chemical fertilizer would decrease the productivity of N absorbed from chemical fertilizer (Figure 5a). No significant relationship was found between the productivity of P absorbed from chemical fertilizer and P absorbed from sources other than chemical fertilizer, the same as K (Figure 5b,c).

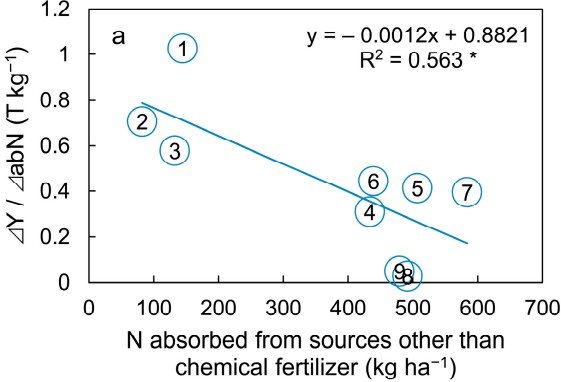

**Figure 5.** *Cont*.

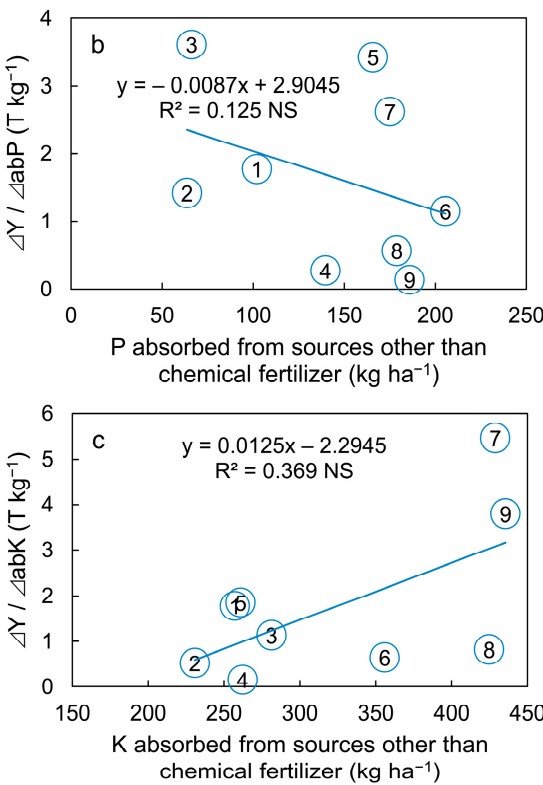

**Figure 5.** Correlation of the productivity of nutrients absorbed from chemical fertilizer with nutrients absorbed from sources other than chemical fertilizer by stem lettuce. ΔY/ΔabN, ΔY/ΔabP, and ΔY/ΔabK represent the productivities of N, P, and K absorbed from chemical fertilizers, respectively. (**a–c**) Numbers in the circles represent the fields for the plot experiments. NS is not significant, * significant at $p < 0.05$.

## 4. Discussion

### 4.1. Mitigating Nutrient Load and Environmental Pollution by Reducing Chemical Fertilizer Application

Excessive chemical fertilizer would increase the nutrient load in the agroecosystem and lead to environmental risk [23]. In the areas that received excessive nutrient inputs, reducing chemical fertilizer application would be beneficial economically, agronomically, and environmentally [24]. Meanwhile, maintaining the crop yield should be considered simultaneously. According to the results, in field 1, the P and K fertilizers could be exempted, while the N fertilizer should not be reduced. In fields 2 and 3, N and P fertilizer was needed, but the K fertilizer was not. In other fields, the applying amounts of N, P, and K could decrease to zero (Table 4).

### 4.2. Critical Levels of Soil Nutrient and Rational Duration for Fertilizer Reduction without Yield Loss

The residual nutrient effects have been observed in many regions, such as India [25], the UK [26], Morocco [27], Japan [7], and China [6]. Our previous studies also indicated that the overuse of chemical fertilizer resulted in N, P, and K accumulation in vegetable field soil in the research area, and different chemical fertilizer applications caused different residual degrees of nutrients [9,18]. Under the effect of residual nutrients, there should be a critical level of available soil N, P, or K sufficient for vegetable growth without additional fertilization. However, the literature that concluded how much the soil nutrient contents met the demands of plastic greenhouse crop production was not sufficient [3]. According to the result of this study, the critical levels of nitrate, Olsen–P, and exchangeable K were 173.3 (field 4), 45.8 (field 1), and 93.1 mg kg$^{-1}$ (field 3), respectively (Tables 2 and 4). As there was a relatively large interval of soil nitrate between field 3 and field 4 (69.4 mg kg$^{-1}$), the more accurately critical level of nitrate may lie between 103.9 and 173.3 mg kg$^{-1}$. However,

the present result should still provide a fertilizer reduction suggestion in the regions where soil residual nitrate is higher than 173.3 mg kg$^{-1}$. A study has summarized data from nearly 100 pieces of literature and has indicated that the critical content of soil Olsen–P for leafy vegetables is 46.0 mg kg$^{-1}$, while it lacks the support from field experiments [28]. The present study provided field experimental data (45.8 mg kg$^{-1}$). Compared to nitrate, K$^+$ is more easily adsorbed by negatively charged soil colloids and relatively hard to lose from soils [29]. This should be the reason why all the soils in the present study showed a relatively high K content. Therefore, the more accurately critical level of exchangeable K may be lower than 93.1 mg kg$^{-1}$. At least this study indicates that there should be a certain room for K fertilizer reduction in the regions where soil exchangeable K is higher than 93.1 mg kg$^{-1}$. In the study area, urea was a common N fertilizer. Therefore, both decline in soil pH and the nitrate accumulation should result from nitrification (Figure 2b) [30]. Meanwhile, soil nitrate from previously excessive fertilization was responsible for soil salinization (Figure 2a). In addition, the significant correlations indicated that EC could evaluate soil nitrate. In this study, the critical level of nitrate, 173.3 mg kg$^{-1}$, corresponded to the EC with the value of 81.5 ms m$^{-1}$ (Figure 2a), meaning farmers could reduce N fertilizer application when soil EC is greater than this value.

### 4.3. Nutrient Balances under Fertilizer Reduction

At the average application levels in the research area (100% N, P, and K), the input–output balances of N, P, and K were 49.6–242.4 (averaging 144.7), −9.1–103.8 (averaging 46.3), and 116.0–308.5 (averaging 225.6) kg ha$^{-1}$ y$^{-1}$, respectively, suggesting that abundant nutrient was surplus, which was unsustainable (Tables 1 and 3). After fertilizer reduction, the input–output balances of N, P, and K were negative. Although the yields were not significantly decreased by at least four crops (one year) in most fields (Table 4), the negative balance would deplete the soil nutrient pools and create soil fertility risk after long–term cultivation [31,32].

Plastic greenhouse vegetables have a faster growth rate than field crops, easily forming a depletion zone of rhizosphere nutrients. Thus, enhancing the movement of soil nutrients, such as improving transpiration to accelerate the nutrient transfer through soil water flow, effectively eliminates the zone, possibly extending the duration of fertilizer reduction [33]. The limitation of this study is the relatively short experimental duration. To determine the rational duration of fertilizer reduction, a further long–term study combined with soil nutrient dynamic testing needs to be performed.

### 4.4. Effect of Residual Nutrients in Soil on Nutrient Absorption and Vegetable Yield

Nutrients absorbed by crops from sources other than chemical fertilizer includes soil nutrients, nutrients in applied crop residues and manure, nutrients from irrigation, and natural processes, such as atmospheric deposition, and biologically fixed N [34,35]. Many studies have reported that there is a competitive effect of residual N on synthetic N, that is, the more crop N is taken up from sources other than the applied synthetic N, and the less synthetic N absorbed by crops [5,36,37]. This study also observed such a competitive relationship (Figures 3a and 4a).

Low nutrient recovery efficiency is a common concern in agricultural management, usually indicating fertilizer waste and environmental harm [4,38–40]. Low nutrient recovery efficiency was also observed in this study. In plots experiments conducted in one field, due to the previously similar cropping management by the same farmer, soils used for different treatments may show a similar nutrient content. Thus, a higher nutrient application rate may result in a lower nutrient recovery efficiency, indicating that reducing the fertilizer input is an effective method to improve nutrient recovery efficiency [41]. However, for a large–scale study, fertilizer management should be different in different fields, inducing various residual nutrients in the soil. When determining the factors affecting the nutrient recovery efficiency, in this case, the residual nutrient effect should be considered [12,42]. For example, in this study, REN values in fields 4–9 were lower than those in fields 1–3

due to more residual nitrate in fields 4–9 (Figure 4a). Many studies have suggested a unified standard of fertilization to improve the nutrient recovery efficiency in vegetable production, which is convenient to extend [28,40,41]. However, for precision fertilization, the recommended fertilization should be detailed based on considering various residual nutrient statuses in different regions.

The "partial factor productivity" (PFP) has been widely used to estimate the effect of a single fertilizer application, typically N, P, or K, on the crop yield [37,42–45]. However, the calculation of PFP cannot ascertain the effect of nutrients absorbed from fertilizer on the yield. Thus, this study calculated the productivity of nutrients absorbed from chemical fertilizer that could reflect the response of the vegetable yield to nutrient uptake from chemical fertilizer. High N absorbed from sources other than chemical fertilizer, namely, residual N in the soil, resulted in a weak yield response to N absorbed from chemical fertilizer (Figure 5a), which coincided with another study [46].

### 4.5. Affecting Mechanism of Fertilizer Reduction on Vegetable Yield

The previous studies found that a history of relatively low N fertilization resulted in less residual nitrate in fields 1–3 than that in fields 4–9 [9,18]; therefore, the REN in fields 1–3 showed high values. The amount of N applied to each field was equal; thus, high REN implied much N uptake from chemical fertilizer. Consequently, under no N fertilizer treatment, the absorbed N from chemical fertilizer in fields 1–3 more easily decreased than in fields 4–9. Additionally, the productivities of N absorbed from chemical fertilizer in fields 1–3 under 100% application of N fertilizer were high (Figure 5a), suggesting that decreasing the N absorbed from chemical fertilizer for one unit would induce much yield loss. These should be why reducing N application resulted in yield loss in fields 1–3. In fields 4–9, owing to the abundantly residual nitrate, the REN and productivities of N absorbed from chemical fertilizer showed low values. Thus, no N fertilizer treatment did not affect the stem lettuce yield. In summary, residual nitrate in soil was the limiting factor of yield under chemical fertilizer reduction. Therefore, determining soil nitrate was essential in guiding fertilization when residual nitrate was various.

### 5. Conclusions

Under the effect of residual nutrients, reducing fertilizer application without yield loss was feasible when soil nitrate, Olsen–P, and exchangeable K were $\geq$173.3, 40.3, and 93.1 mg kg$^{-1}$, respectively, which might provide guidance for fertilizer reduction in the regions that received residual nutrient from previous fertilization. Residual soil nitrate was the main factor that greatly influenced vegetable yield under fertilizer reduction. The concrete mechanism was that N absorbed from sources other than chemical fertilizer by vegetables was mainly from soil nitrate, which was negatively correlated with the REN and the productivity of N absorbed from chemical fertilizer, respectively. Soil EC value could be used to characterize the nitrate content, which is helpful for reducing N fertilizer without yield loss when residual nitrate in the soil was various.

**Author Contributions:** Conceptualization, N.Z. and Y.W.; methodology, Y.W.; software, Y.C.; validation, J.W.; formal analysis, W.Y.; investigation, N.Z.; data curation, N.Z.; writing—original draft preparation, N.Z.; writing—review and editing, Y.W.; visualization, Y.W.; supervision, Y.W.; project administration, Y.W.; funding acquisition, Y.W. All authors have read and agreed to the published version of the manuscript.

**Funding:** This work was supported by the Natural Science Foundation of Anhui Province (1708085QD88, 2008085QD162), the University Synergy Innovation Program of Anhui Province (GXXT-2020-075), the Foundation of Anhui Laboratory of Molecule-Based Materials (fzj19012) and Anhui Provincial Key Laboratory of Nutrient Recycling, the Anhui Provincial Foundation for Overseas Students.

**Institutional Review Board Statement:** Not applicable.

**Data Availability Statement:** When requested, the authors will make available all data used in this study.

**Conflicts of Interest:** The authors declare no conflict of interest.

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
