# Peer review of "Reducing Chemical Fertilizer Application in Greenhouse Vegetable Cultivation under Different Residual Levels of Nutrient"

_agriculture, doi:10.3390/agriculture13061174_

Round 1
Reviewer 1 Report
Reviewer comments on Reducing chemical fertilizer application in greenhouse vegetable cultivation under different residual levels of nutrient. The paper is well-written and clearly presented results. However, following remarks should be considered before further processing.
1. L89. botanical name should be italic.
2. Author mention potassium chloride in text but also mention K2O in text in line number 103 of manuscript. It should be KCl in place of K2O. Elemental names are not correct!
3. Author should reframe sub heading of material method line number 113 of manuscript such as Analyses of soil and vegetable nutrient.
4. Author mentioned “The five soil subsamples were thoroughly mixed and were separated into 116 two parts after sieving (< 5 mm), one of which was utilized to analyze nitrate content (the 117 main form of N absorbed by vegetables) and the other was air–dried to determine soil 118 total N (TN), total carbon (TC), available P and K, pH, and electric conductivity (EC)”. Author should give detailed about five soil subsamples.
5. Author mentioned the subsample point area was 1 m2 in 129-line number of manuscript. Author should recheck it.
6. Author should rephrase the sentence Additionally; no yield loss was observed 193
in other fields under no P fertilization treatments in line number 193-194 of manuscript.
7. Author mention subheading in non-italic and also in italic form. The sub heading of manuscript should be same formats.
8. Reference number 29, 43and 45 are different format. Reference should be in same format
Minor improvment is required!
Reviewer 2 Report
This study explores the possibility of reducing chemical fertilizer use in greenhouse vegetable cultivation without yield loss. Nine plot experiments were conducted in fields with varying residual nutrient levels, highlighting the importance of considering residual nutrient levels in recommending fertilizer reduction for sustainable agriculture and environmental safety.
However, despite the involvement of soil physical and chemical properties in crop absorption, this paper does not describe the basic soil properties such as clay content, CEC, base saturation, exchangeable base content, Alo, and Feo. Meteorology is also a significant factor in crop production, but it is not mentioned in the paper. Therefore, while the results obtained are useful for local agriculture, they are insufficient for publication in an international journal. The argument of the article should also be made clearer. The main point that the summary wants to convey is not clear.
The title of Table 3 should be “nitrogen absorbed” not “Nutrient absorbed”.
There are no major problems regarding the quality of the English language.
Reviewer 3 Report
First of all, the paper is well written, and results are interesting and well discussed. I recommend this paper, with few minor edits and suggestions.
1. Line 129: shoudl be m2
2. A mass balance depicting input and output fertilizers in the soil can be very helpful and give more insights.
3. In the conclusion section, the broader implications of the results of this study need to be specified, rather than sticking to only the study area. What would you specify to the farmer for his decision-making after analyzing the soil nitrate concentrations?
4. The limitations of this study should also be clearly mentioned along with future research needs.
The English of the paper is really good. However, in some places, the sentences can be more compact (for example, lines 300-302, can be combined into one sentence). Similar instances in the paper need to be rectified.
Round 2
Reviewer 2 Report
Thank you for your revised manuscript. I think the revisions are good, but after re-reading it, I noticed the following points. I am afraid I did not mention it in my previous comment.
For nitrogen, the threshold cannot be determined to be 173.3 because it can be divided into groups 1 to 3 and 4 to 9. In other words, the threshold must lie somewhere between 103.9 and 173.3.
For phosphate, the results are reasonable. However, they are largely consistent with existing reports, and there is no originality.
For potassium, Figures 3 and 4 seem to indicate that there is no clear threshold.
Based on these results, I regret to say that I cannot recommend this paper for publication in agriculture.
If I have misunderstood, please send me your comments.
I think the English quality is enough.
Author Response
Please see the attachment, thank you!
